# The spatiotemporal distribution of Japanese Encephalitis cases in Yunnan Province, China, from 2007 to 2017

Xianghua Mao[1,2], Hongning Zhou[2]*

**1** Yunnan Provincial Center of Arbovirus Research, Pu'er, Yunnan, China, **2** Yunnan Institute of Parasitic Diseases, Pu'er, Yunnan, China

* zhouhn66@163.com

## Abstract

### Background

Japanese encephalitis (JE) is a vector-borne disease with a high prevalence in Yunnan Province, China. However, there has been a lack of a JE epidemic systematic analysis, which is urgently needed to guide control and prevention efforts.

### Methods

This study explored and described the spatiotemporal distribution of JE cases observed among two different age groups in Yunnan Province from 2007 to 2017. The epidemiological features and spatial features were analyzed according to basic statistics, ArcGIS software (version 9.3; ESRI, Redlands, CA) and SPSS software (version 20; IBM Corp., Armonk, New York).

### Results

Overall, the whole province had a high incidence of JE. The annual incidence rates in 2007 and 2017 were 1.668/100,000 and 0.158/100,000, respectively. The annual mortality was under 0.095/100,000 for these years. Although the whole province was in danger of JE, the Diqing autonomous prefecture and the Lijiang autonomous prefecture had no JE cases recorded for over 10 years. The JE cases were reported by hospitals located in 60 counties of 14 municipalities. The top ten areas with the most JE cases were Kunming City, Zhaotong City, Jinghong City, Wenshan City, Mangshi City, Pu'er City, Baoshan City, Dali City, Chuxiong City, and Gejiu City. The incidence declined smoothly, with a peak occurring from June to September, which accounted for 96.1% of the total cases. Children whose age was equal or less than 10 years old (LEQ10) still maintained a high frequency of JEV infection, and a large number of cases were reported in August, despite the Expanded Program on Immunization (EPI), which was established in April 2008. There was no difference in the quantity of cases between the two groups ($t = -0.411$, $P > 0.05$); additionally, the number of JE cases among patients LEQ10 were significantly greater than those among patients older than 10 years (GTR10). Further analysis using local indicators of spatial association (LISA) revealed

**Data Availability Statement:** All relevant data are within the paper and its Supporting Information files. The JE data used in this study were obtained from the China Information System for Diseases

Control and Prevention (available at http://www.phsciencedata.cn/).

**Funding:** This work was supported by grants from National Natural Science Foundation of China (U1602223), and Major Science and Technology Projects in Yunnan Province (2017ZF007).The funders had no role in study design, data collection and analysis,decision to publish, or preparation of the manuscript.

**Competing interests:** The authors have declared that no competing interests exist.

that the distribution of JE exhibited a high-high cluster characteristic ($Z = 2.06$, $P<0.05$), which showed that Jinghong City, Guangnan County, Yanshan County, Funing County, and Mengzi City were hot spots for the JE epidemic.

## Conclusions

Although the EPI was established in 2008 and the incidence of JE declined smoothly in Yunnan Province, there was no difference in the number of cases between the two age groups, which reveals that the EPI has been conducted with a low level success. In the context of limited vaccine supply capacity, we should strengthen the implementation of the children's immunization program before strengthening other immunization programs.

## Introduction

Japanese encephalitis (JE) is a mosquito-borne disease caused by a virus that belongs to the family Flaviviridae, genus Flavivirus [1, 2], which causes approximately 30,000 to 50,000 cases and 10,000 to 15,000 deaths in Asian countries annually [3], with an overall incidence of 1.8 per 100,000 [4]. During an epidemic, approximately 0.1–4% of infected individuals develop clinically apparent encephalitis [5]. However, these figures are probably underestimated [6]. An estimated 3 billion persons live in countries where the JE virus (JEV) is endemic [7]. In these areas, the case fatality rate is 20–30%, and 30–40% of survivors suffer from permanent neurological sequelae [8]. Previous studies from India have ascertained that for every case of symptomatic JE, 200 patients suffer from asymptomatic or subclinical disease [9]. JE was first described in 1871, and the first recognized epidemic occurred in Japan in 1924 [10–12]. The prototype Nakayama JEV strain was isolated from a postmortem human brain in Tokyo, Japan, in 1935 [13, 14]. Phylogenetic analysis of JEV strains revealed that JEVs can be divided into four genotypes using the prM gene [15] and five genotypes according to the E gene or full-length genome [16, 17]. Genetic studies suggest that JEV originated from an ancestral virus in the area of the Malay Archipelago, after which the virus evolved into different genotypes (I–IV) and spread across Asia. Until recently, most of the strains of JEV at the origin of major epidemics in the South, East, and Southeast Asia regions belonged to genotype III [18]. However, a shift in prevalence from JEV genotype III to JEV genotype I has been observed in several Asian countries [19–21], although an evolutionary study has demonstrated that three genotypes (I, III, V) were confirmed to be co-circulating in China in both high and low prevalence areas [22]. All JEV strains belong to a single serotype, as evident from an epidemiological observation of the absence of secondary encephalitis [23]. The main vector of JEV transmission is *Culex tritaeniorhynchus* [24], although it can be transmitted with vector-free [5]. In Australia [25, 26], *Culex annulirostris* was identified as the main vector for JEV transmission. Domestic pigs are considered the major amplifying hosts, and humans are considered dead-end hosts [27, 28].

JE is one of the ongoing epidemics in Yunnan Province, China. Although many measures have been taken in recent years, both the accumulated experience and international cooperation for JE control, especially in the management of the program and the analysis of data, are either moving backward or closed. Therefore, we urgently need to analyze the data collected from a primary survey to inform a reliable and effective strategy for the termination of JE. Furthermore, we also expect to exchange views with the international community.

## Materials and methods

### Study region

Yunnan Province is located on the southwestern border of China between 21–29° north latitude and 97–106° east longitude. The borders are adjacent to Guizhou and Guangxi in the East, Sichuan in the North, Tibet in the Northwest, Myanmar in the West, and Laos and Vietnam in the South, and the province occupies a total area of 394,000 square kilometers. While the climate is largely subtropical and tropical monsoon, there is a plateau climate in the Northwest. By the end of 2017, Yunnan Province had jurisdiction over 129 counties or districts that belong to eight provincial municipalities or eight autonomous prefectures.

### Data collection and management

China has a relevant reporting system of epidemics and strong support of the network for data collection. We obtained the database of JE cases from the China Information System for Diseases Control and Prevention (CISDCP). A list of study regions was created for data collection, and the inclusion and exclusion criteria were provided. The factors collected included sex, age, date of onset, occupational distribution, case classification, and reporting areas. Personal identifying factors, such as name, address and clinical information, were removed. Individuals were divided into two groups by age to analyze the differences between them; one group was LEQ10 and the other group was GTR10. Then, we collected demographic information from the national statistical bulletin. The data analyzed in this study included the total number of outbreaks, occupations of the patient, time of symptom onset, reporting area and reporting time. The use of clinical symptoms or medical information was forbidden in this study. All data were analyzed by the ArcGIS software (version 9.3; ESRI, Redlands, CA) and SPSS software (version 20; IBM Corp., Armonk, New York) automatically.

### Analysis of the epidemiological characteristics

The annual incidence and annual mortality of JE were displayed by wave patterns, with a scale of 1 to 100,000. The seasonal pattern was displayed in a histogram based on the monthly cases from 2007 to 2017. The t-test was used to analyze differences among JE cases by age, which was divided into two groups: one group was LEQ10, and the other group was GTR10. A ratio was used to reveal the distribution of males and females. The distributions of the JE cases by epidemiological parameters, including sex, age and occupation, were calculated according to basic statistics.

### Spatial cluster analysis

All data from the JE cases were plotted with ArcGIS software (version 9.3; ESRI, Redlands, CA). The geographical distribution of JE cases was divided into five levels: 0, 1–3, 4–10, 11–30 and above 30 cases. All data were displayed on GIS maps at the county level or municipal level. The local indicators of spatial association (LISA) were used to analyze the distribution of JE, and the Z-score, P-value and local Moran's I coefficient were calculated. A local Moran's I coefficient >0 indicates that the data have a positive spatial correlation. In contrast, a local Moran's I coefficient <0 indicates that the data have a negative spatial correlation, and a local Moran's I coefficient = 0 indicates that the data are random. In addition, the Z-score and P-value were utilized for data analysis. When the Z-score was > 1.96 ($P<0.05$), the distribution of JE cases was clustered, which indicated that the surrounding features had similar values (high-high or low-low), and when the Z-score was < -1.96 ($P<0.05$), the distribution of JE cases was dispersed, which indicates that the surrounding features had a significant spatial

outlier (high-low or low-high). If the Z-score was between -1.96 and +1.96 and the P-value was > 0.05, the data were random.

## Results

### Epidemiological features of JE in Yunnan Province

**Incidence and mortality.** A total of 3038 cases were reported from 2007 to 2017, with an annual incidence between 0.158/100,000 and 1.668/100,000 and an annual mortality between 0.004/100,000 and 0.095/100,000. While the annual incidence and mortality in 2007 were 1.668/100,000 and 0.095/100,000, respectively, they decreased to 0.158/100,000 and 0.004/ 100,000 in 2017, respectively, which suggested a downward trend from 2007 to 2017 (Fig 1).

**Sex.** Among the 3038 JE cases reported from 2007 to 2017, 1802 were male and 1236 were female, reflecting a male to female ratio of 1.46:1.

**Seasonal pattern.** The distribution of cases shows a clear seasonal pattern, which had the highest wave from June to September with a peak in August, whether the incidence for a given year was high (2007) or low (2017). A total of 96.1% of the total cases occurred from June to September, while 72.6% occurred in July and August (Fig 2).

**Distribution by age.** Among both age groups, the median age was 10 years old, with a minimum age of 0.1 years old and a maximum age of 86 years old. Of the cases that had been reported from 2007 to 2017, 1625 were LEQ10, and 1413 were GTR10, reflecting a ratio of 1.15:1. The proportion of JE patients whose age was GTR10 was 46.5%, while the proportion of JE patients whose age was LEQ10 was 53.5%. The highest number of cases was reported in 2007 in both age groups.

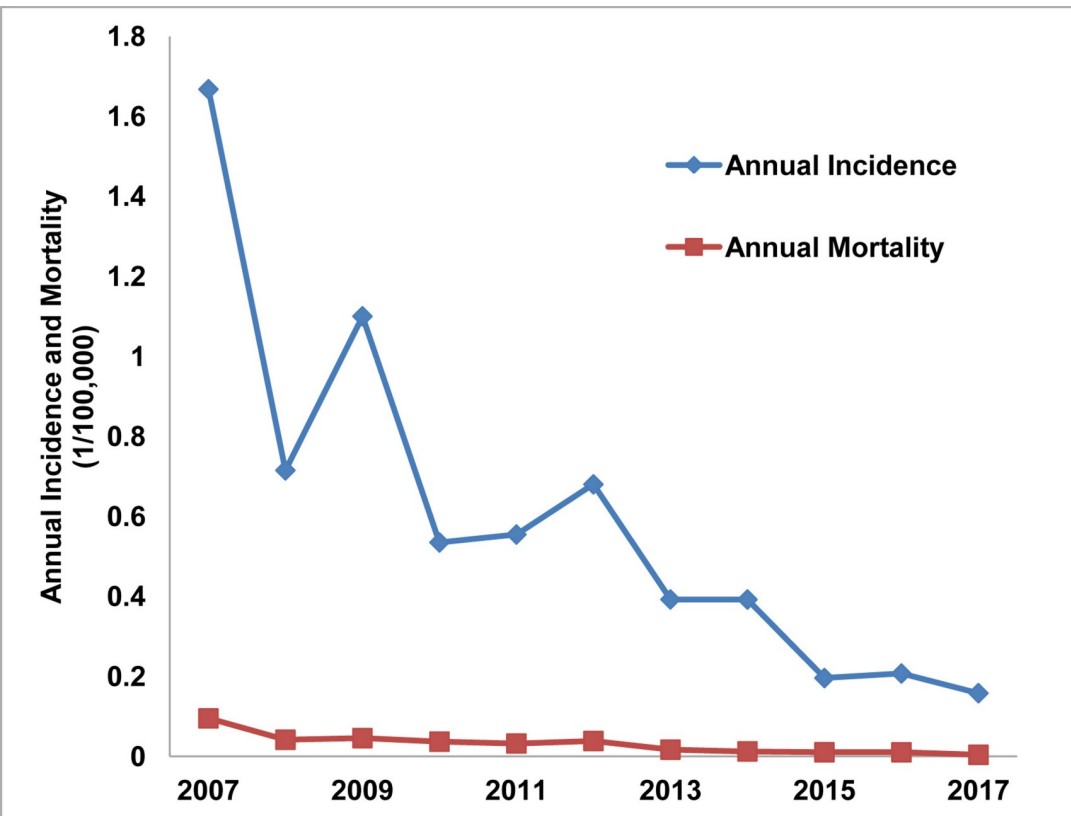

**Fig 1. Annual incidence and mortality of JE in Yunnan Province, China, 2007–2017.**

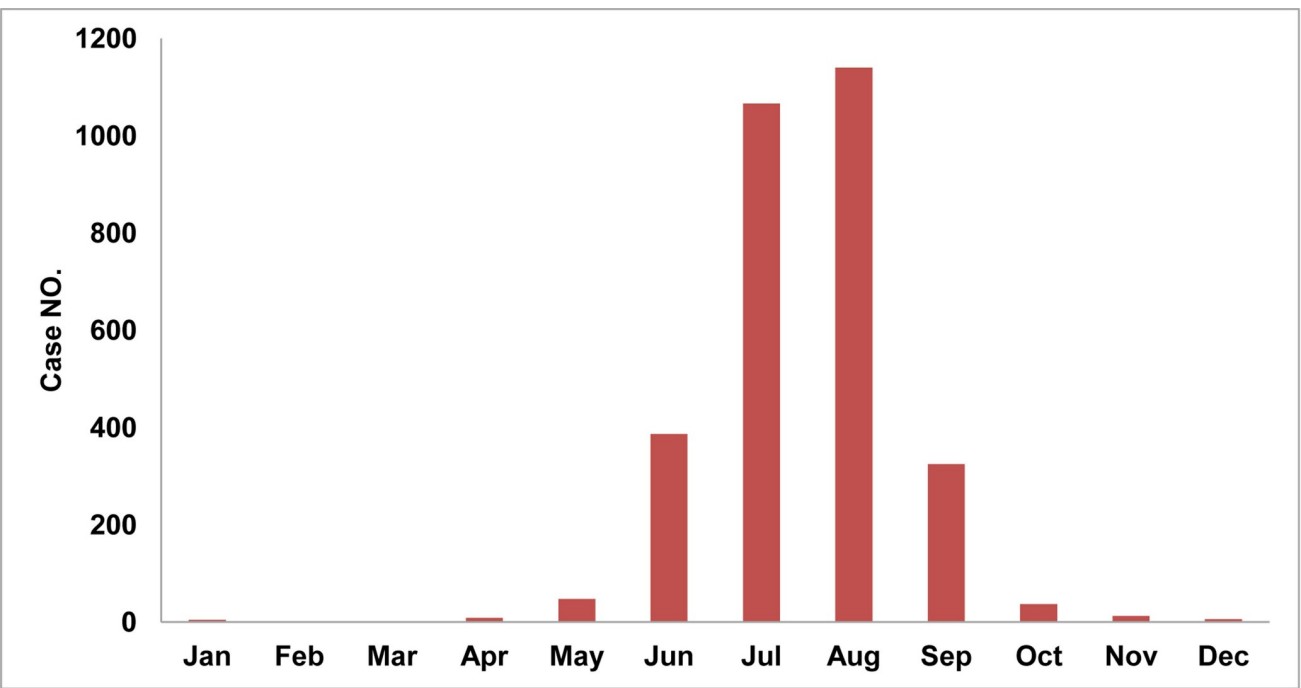

**Fig 2. Monthly cumulative cases in Yunnan Province, China, 2007–2017.**

**Occupational distribution.** Children and students accounted for a large portion of cases (68.5% of the total). Additionally, the JE incidence was highest among children, accounting for 35.6% of the total. Among the other cases, medical staff, merchants, farmers and urban residents accounted for 31.5% of the total occupations.

**Geographical distribution.** The geographical distribution of JE changed every year. Since 2007, the cases exhibited a downward trend until 2011. In 2011, the region of Zhaoyang, which declined in 2010, increased notably. In 2007, there were 60 counties or districts with reported cases, which dropped to 20 counties by the end of 2017, reflecting a decline of 66.7%. From 2015, the number of cases decreased significantly, dropping to less than 100 cases in both age groups. In the past decade, the top ten areas according to case reporting were Kunming City, Zhaotong City, Jinghong City, Wenshan City, Mangshi City, Pu'er City, Baoshan City, Dali City, Chuxiong City, and Gejiu City. In 2017, the top five areas in terms of cases reported were Kunming City, Gejiu City, Pu'er City, Jinghong City, and Mengzi City (Fig 3).

From 2007 to 2017, 82 counties of 14 municipalities reported JE cases; however, the Diqing autonomous prefecture and the Lijiang autonomous prefecture were free of JE and had no reports of JE for over 10 years. The top five municipalities according to the number of reported cases were Kunming City, Zhaotong City, Wenshan City, Pu'er City, and the Xishuangbanna autonomous prefecture. In terms of the two age groups, the geographical distribution of JE was different between them. In the group aged GTR10, Kunming City, Zhaotong City, Pu'er City, the Xishuangbanna autonomous prefecture and the Dehong autonomous prefecture were ranked among the top five areas in case reporting, while the Wenshan autonomous prefecture, Zhaotong City, Kunming City, the Xishuangbanna autonomous prefecture and the Dehong autonomous prefecture were ranked among the top five in the group aged LEQ10 (Fig 4).

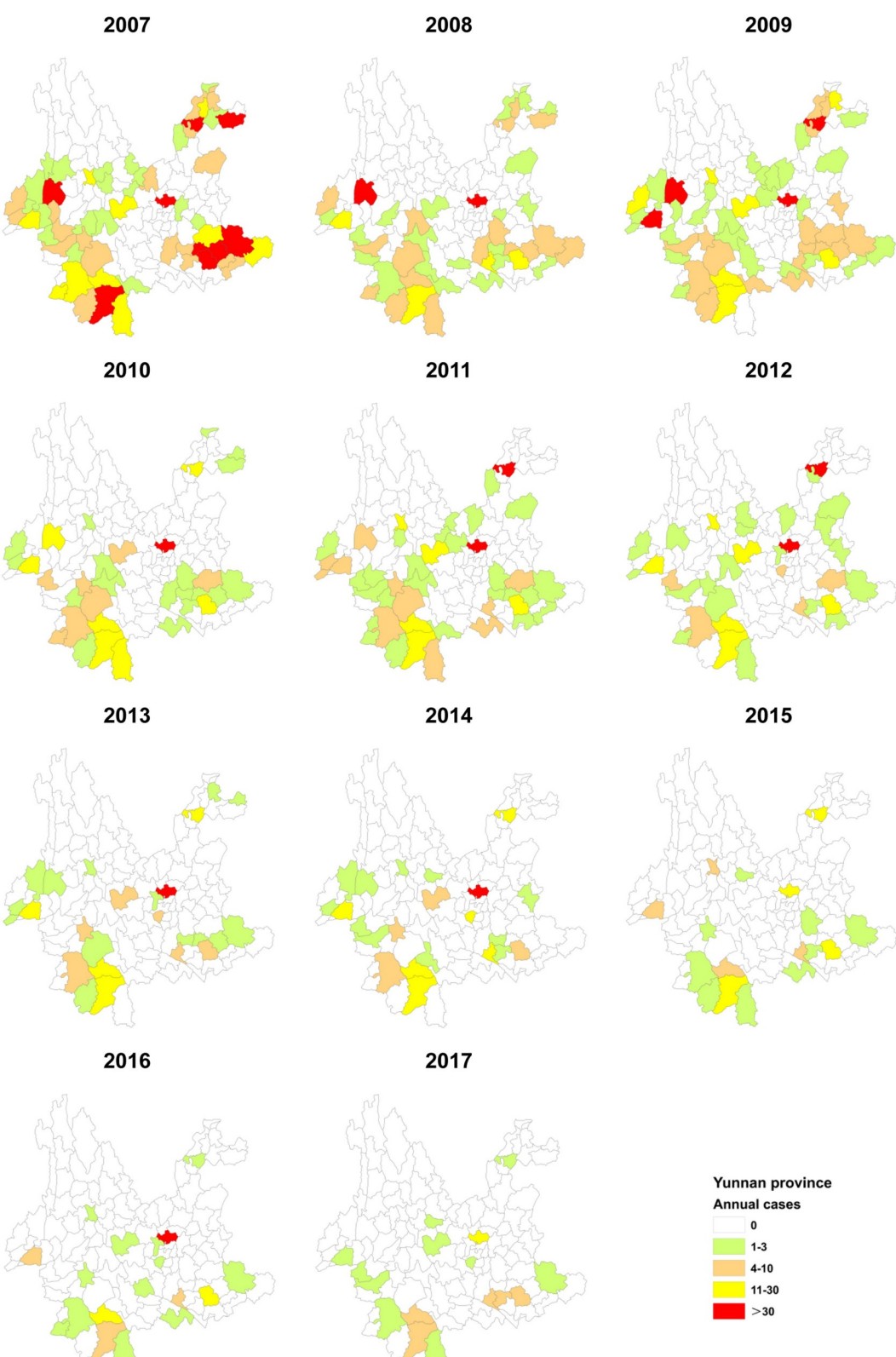

**Fig 3. Spatiotemporal distribution of JE in Yunnan Province, China, 2007–2017.** The cases of JE plotted on the map were derived from the crude data at the county level.

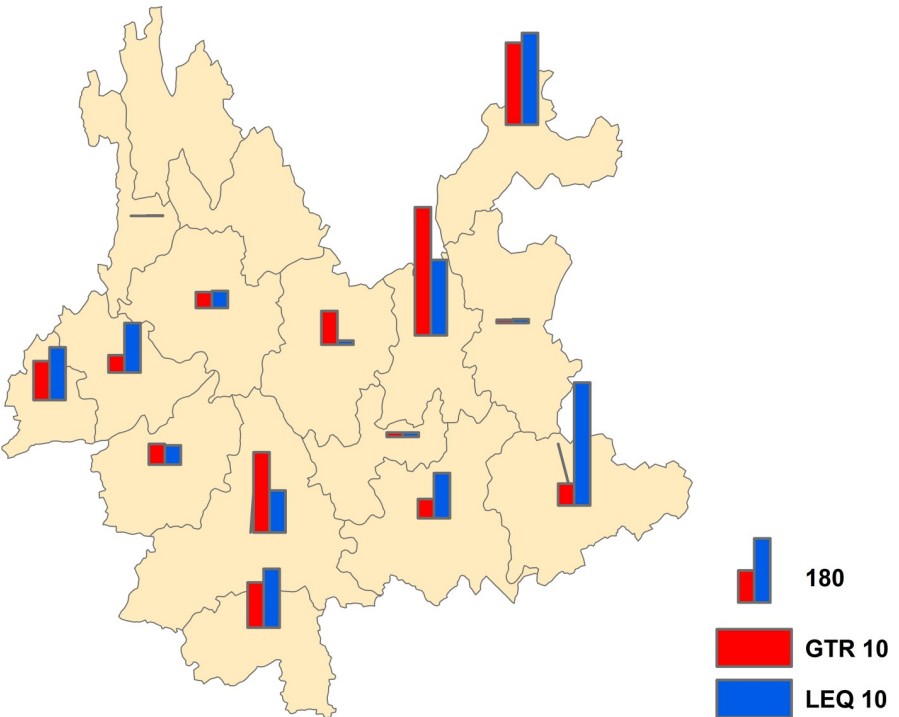

**Fig 4. Geographical distribution of the two age groups in Yunnan Province, China, 2007–2017.** The cases of JE plotted on the map were derived from the crude data at the municipal level.

### Spatial features of JE in Yunnan Province

**Dispersion and clustering.**   The cluster analysis showed that there was a significant agglomeration. In 2007, both Kunming City and Longyang District had high-low cluster characteristics, while Jinghong City, Guangnan County, Yanshan County, and Funing County had high-high cluster characteristics (Z = 2.06, P<0.05). In 2017, although Kunming City had high-low cluster characteristics and both Mengzi City and Jinghong City had high-high cluster characteristics, the clustering was not obvious, which suggested that the JE reporting was random (Fig 5).

**Statistical analysis.**   Among the 3038 cases that were reported from 2007 to 2017, there were 1625 patients aged LEQ10 and 1413 patients aged GTR10. The statistical analysis showed that there was no significant difference between these two age groups (*t* = -0.411, *P* = 0.686). Furthermore, the number of patients aged LEQ10 was greater than the number of patients aged GTR10 (1.15:1).

## Discussion

Recently, there has been a great amount of support for vaccinating adults against JE. Supporters believe that it is urgent to carry out vaccination for adults based on the number of adult patients and the age of onset. Their claim is unrealistic and unsustainable for all adults to be vaccinated, at least in Yunnan Province, which does not have ample facilities. In fact, the in-depth analysis of epidemiological JE data is limited and it has been erroneously concluded that adults need to be vaccinated immediately. Other provinces of China believe that the strategy of vaccination for adults would not respond to the actual situation in Yunnan Province.

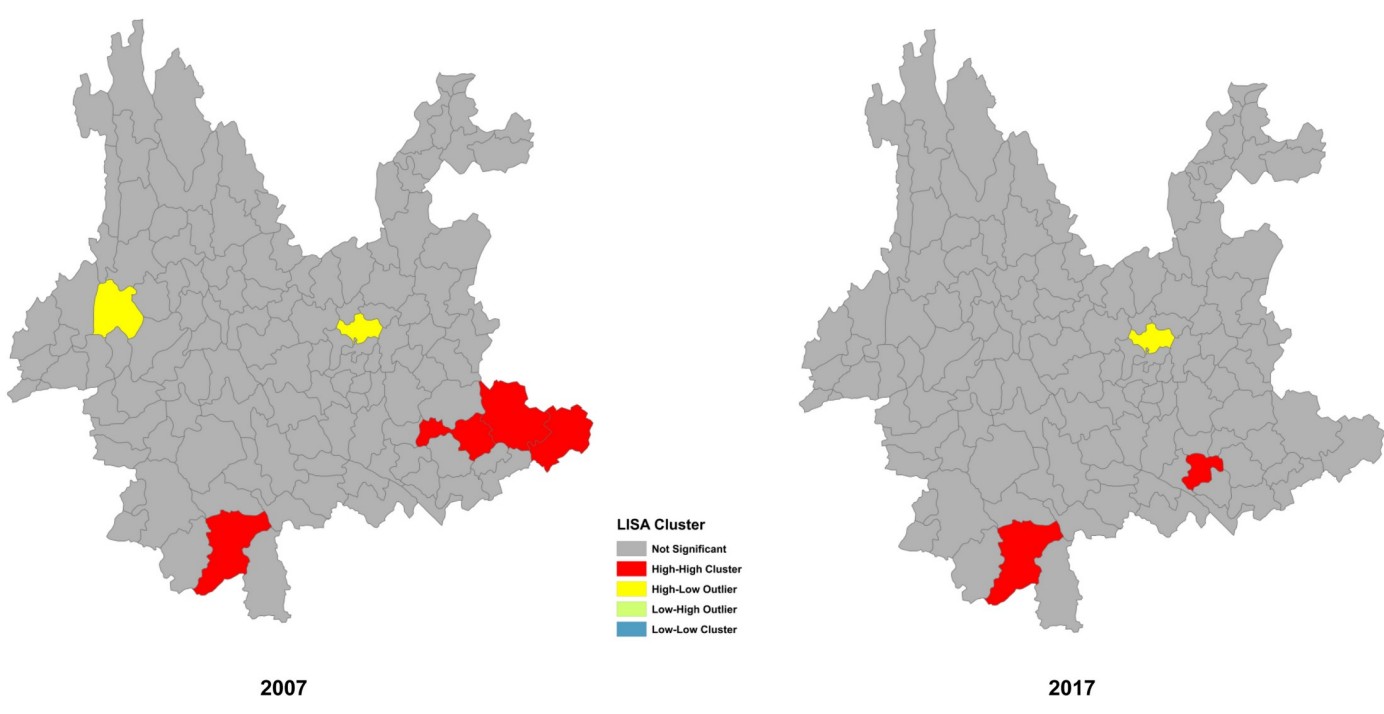

**Fig 5. Dispersion and clustering of JE reporting.**

Although the incidence and mortality of JE have shown a downtrend in Yunnan Province, China, the risk of an epidemic or an outbreak is still high. The incidence and mortality of JE in 2007 was 1.668/100,000 and 0.095/100,000, respectively, and they decreased to 0.158/100,000 and 0.004/100,000, respectively, in 2017. In recent years, we established a strong system for epidemic reporting, and the system has been running smoothly. However, most cases of JE are asymptomatic and go unreported, which makes the spatial distribution of the virus difficult to estimate [29]. For example, the reporting of cases depends on the quality of the health information systems and the ability to clinically and serologically diagnose the disease [10]. Meanwhile, these data were from a passive surveillance system, which means that there may have been underreporting of JE because some cases had subclinical symptoms [10, 30]. In a study that was reported in 2009, the infection rate of *Culex tritaeniorhynchus* was 13.2% [31], indicating a high level of virus activity. In Tibet, during a JE epidemic, the overall seroprevalence of JEV IgM in pigs was 5.1% [32]. Furthermore, JE epidemics have occurred along the border of Yunnan Province [33, 34]. In addition, geographic changes; agricultural changes; ecological factors; birds; climate changes; changing risks from rural exposures; changing profiles of international travelers; peri-urban risk; personal factors and JE risk; non-mosquito vector transmission in pigs; the incidence of JEV infections in febrile travelers; JEV infection without encephalitis; changes in trip duration for JE cases; and changes in JE vaccine utilization can also cause an outbreak or epidemic [35]. This study suggested that the risk of another JE epidemic occurring is still high and that prevention efforts in Yunnan Province should not be neglected.

JE is widely distributed in Yunnan Province except for in the Lijiang autonomous prefecture and the Diqing autonomous prefecture. For the years studied, the prevention of JE in Yunnan Province has been steadily carried out, and human cases have been found or reported in most areas. The Lijiang and Diqing autonomous prefectures, however, had no cases reported in over 10 years. The JE cases were not randomly distributed [36], they also show the

same characteristics in Yunnan Province, especially in the areas of Jinghong City, Guangnan County, Yanshan County, Funing County, and Mengzi City, which are hot spots for the JE epidemic. Usually, cases in Yunnan Province are treated by nearby hospitals; however, some of the cases reported in an area were not acquired in that area; sometimes the number of cases only reflects the strong diagnostic testing capacity of the area [10]. The number of treatments by hospitals, however, is roughly equal to the actual number of cases because of the special expense reimbursement system and process in China. Therefore, the incidence of JE showed a downward trend, and the epidemic range is becoming increasingly smaller. In the Lijiang and Diqing autonomous prefectures, where there is no JE epidemic, we should carefully investigate whether this is due to their limited facilities [10].

Children should be given priority with regard to the JE vaccine. According to a study that monitored JE in Jinan, Yichang, Shijiazhuang, and Guigang, cases can be found in sentinel hospitals and non-sentinel hospitals [30]. People also have increased awareness of the prevention of JE [37]. These results provide a prerequisite for the implementation of immunization. The benefits of vaccination are also obvious from a study conducted by Guizhou Province in China, which showed that immunization of 100,000 persons was predicted to save 1.6 billion dollars for the health system and 11.6 billion dollars for society [38]. The EPI has been implemented in Yunnan Province for many years. The number of cases in the group aged LEQ10, however, was still at a high level. The results indicate that there was no difference in the number of cases between the two age groups. In addition, the proportion of JE cases among LEQ10 patients was still significantly greater than that among GTR10 patients, which indicates that the implementation of our EPI must be strengthened. In the current study, we found that the EPI has not been implemented very well in some regions, and in a few regions, there was also a shortage of vaccines. Because of the limited supply of vaccines and these constraints, the vaccination of all adults against JE is unrealistic and unsustainable in Yunnan Province. Further cluster analysis revealed that five areas were hot spots for JE cases, and these regions should draw up similar strategies for JE prevention. Furthermore, the whole province should strengthen the vaccination of children as the first option, before vaccinating adults, due to the low supply of the vaccine. For the prevention of JE among adults, more environmental modifications should be adopted, rather than an unrealistic promotion of vaccines with limited availability. The core mission is to provide high-quality and affordable vaccines. At the same time, monitoring and data analysis should be strengthened more than blind work and unrealistic slogans.

## Supporting information

**S1 Data. Spatiotemporal distribution and geographical distribution.**
(XLS)

**S2 Data. Annual incidence and mortality.**
(XLSX)

**S3 Data. Monthly cumulative cases.**
(XLSX)

## Author Contributions

**Conceptualization:** Xianghua Mao.

**Data curation:** Xianghua Mao.

**Formal analysis:** Xianghua Mao.

**Funding acquisition:** Hongning Zhou.

**Investigation:** Xianghua Mao.

**Methodology:** Xianghua Mao.

**Project administration:** Xianghua Mao.

**Resources:** Xianghua Mao.

**Software:** Xianghua Mao.

**Supervision:** Hongning Zhou.

**Validation:** Xianghua Mao.

**Visualization:** Xianghua Mao.

**Writing – original draft:** Xianghua Mao.

**Writing – review & editing:** Xianghua Mao, Hongning Zhou.

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
