## [Decision Letter · Decision Letter 0]

23 Jan 2020

PONE-D-19-35285

The Spatio-temporal Distribution of Japanese Encephalitis Cases in Yunnan Province, China from 2007 to 2017

PLOS ONE

Dear Mr. Mao,

Thank you very much for submitting your manuscript. While both reviewers appreciated the efforts on the revision, there is one remaining concern on the manuscript’s writing (see the comments from reviewer 2 below).   We therefore ask you to edit the manuscript according to reviewer 2’s recommendation before we can consider your manuscript for acceptance.

We would appreciate receiving your revised manuscript by Mar 08 2020 11:59PM. To enhance the reproducibility of your results, we recommend that if applicable you deposit your laboratory protocols in protocols.io, where a protocol can be assigned its own identifier (DOI) such that it can be cited independently in the future. For instructions see: http://journals.plos.org/plosone/s/submission-guidelines#loc-laboratory-protocols

We look forward to receiving your revised manuscript.

Kind regards,

Tian Wang, PhD

Academic Editor

PLOS ONE

2. Please amend the manuscript submission data (via Edit Submission) to include author Hongning Zhou.

Reviewers' comments:

Reviewer's Responses to Questions

**Comments to the Author**

1. Is the manuscript technically sound, and do the data support the conclusions?

Reviewer #1: Yes

Reviewer #2: Partly

2. Has the statistical analysis been performed appropriately and rigorously? 

Reviewer #1: N/A

Reviewer #2: I Don't Know

3. Have the authors made all data underlying the findings in their manuscript fully available?

Reviewer #1: Yes

Reviewer #2: Yes

4. Is the manuscript presented in an intelligible fashion and written in standard English?

Reviewer #1: Yes

Reviewer #2: No

5. Review Comments to the Author

Reviewer #1: All of my concerns have been adequately addressed .

Reviewer #2: Portions of the manuscript have improved compared to the first submission. Unfortunately, the manuscript still contains numerous grammatical errors throughout, which make it difficult to follow and understand the methods. Additional editing is needed.

6. PLOS authors have the option to publish the peer review history of their article (what does this mean?). If published, this will include your full peer review and any attached files.

Reviewer #1: No

Reviewer #2: No

---

## [Author Response · Author response to Decision Letter 0]

6 Mar 2020

Response: Thank you for the helpful comment. In accordance with the comment, we have asked professionals (American Journal Experts, AJE) to carefully revise the manuscript. Additionally, to make sure the statistical analysis been performed appropriately and rigorously, we also added new supporting information to ensure the conclusions.

---

## [Decision Letter · Decision Letter 1]

16 Mar 2020

PONE-D-19-35285R1

The spatiotemporal distribution of Japanese Encephalitis cases in Yunnan Province, China, from 2007 to 2017

PLOS ONE

Dear Mr. Mao,

Thank you for submitting your revised manuscript to PLOS ONE. After careful consideration, we feel that it has merit but does not fully meet PLOS ONE’s publication criteria as it currently stands. Therefore, we invite you to submit a revised version of the manuscript that addresses the points raised during the review process.

We would appreciate receiving your revised manuscript by Apr 30 2020 11:59PM. To enhance the reproducibility of your results, we recommend that if applicable you deposit your laboratory protocols in protocols.io, where a protocol can be assigned its own identifier (DOI) such that it can be cited independently in the future. For instructions see: http://journals.plos.org/plosone/s/submission-guidelines#loc-laboratory-protocols

We look forward to receiving your revised manuscript.

Kind regards,

Tian Wang, PhD

Academic Editor

PLOS ONE

Reviewers' comments:

Reviewer's Responses to Questions

**Comments to the Author**

1. If the authors have adequately addressed your comments raised in a previous round of review and you feel that this manuscript is now acceptable for publication, you may indicate that here to bypass the “Comments to the Author” section, enter your conflict of interest statement in the “Confidential to Editor” section, and submit your "Accept" recommendation.

Reviewer #2: (No Response)

2. Is the manuscript technically sound, and do the data support the conclusions?

Reviewer #2: Yes

3. Has the statistical analysis been performed appropriately and rigorously? 

Reviewer #2: I Don't Know

4. Have the authors made all data underlying the findings in their manuscript fully available?

Reviewer #2: Yes

5. Is the manuscript presented in an intelligible fashion and written in standard English?

Reviewer #2: Yes

6. Review Comments to the Author

Reviewer #2: The authors have addressed my concerns regarding grammatical errors from the previous draft. This revised draft is much improved and easier to follow. I only have a few minor editing suggestions listed below.

Line 41 a high-high cluster? Should this be “a high cluster characteristic?”

Line 74 Can you explain non-vector transmission or give an example here?

Line 90 Change provide to province.

Lines 105-106 This sentence seems redundant and should be removed, if I am understanding it correctly: “Private information had been completely erased from the original JE dataset; therefore, the data we used for this analyses no longer included names, addresses, etc.” Lines 100-101 already state that “Personal identifying factors, such as name, address and clinical information, were removed.”

Line 109 What software? ArcGIS?

Line 123 Change revealed to displayed.

Lines 125-126 Edit sentence to: A local Moran’s I coefficient >0 indicates that the data have a positive spatial correlation.

Line 128 Edit sentence to: In addition, the Z-score and P-value were utilized for data analysis.

Line 212 Edit sentence to: Recently, there has been a great amount of support…

Line 213 Change quantity to number.

Line 218 Change “respond to” to improve.

Line 219 Edit sentence to: Although the incidence and mortality of JE have shown a downward trend in Yunnan Province…

Line 222 Remove: …which seems to be a downward trend of incidence and mortality. This is redundant.

Line 229 Edit sentence to: In a study that was reported in 2009, the infection rate of Culex tritaeniorhynchus was 13.2% [31], indicating a high level of virus activity.

Line 231 Edit sentence to: In Tibet, during a JE epidemic, the overall seroprevalence…

Line 232 Edit sentence to: Furthermore, JE epidemics have occurred along the border…

Line 238 Edit sentence to: This study suggested that the risk of another JE epidemic occurring is still high and that prevention efforts in Yunnan Province should not be neglected.

Line 242 Change any to human.

Line 248 Change “caused or infected” to acquired.

Line 249 Change discovery to diagnostic testing.

Line 257 Change has to have.

7. PLOS authors have the option to publish the peer review history of their article (what does this mean?). If published, this will include your full peer review and any attached files.

Reviewer #2: No

---

## [Author Response · Author response to Decision Letter 1]

25 Mar 2020

Dear Dr. Wang,

We have received the comments about the manuscript entitled “The spatiotemporal distribution of Japanese Encephalitis cases in Yunnan Province, China, from 2007 to 2017” (Manuscript Number: PONE-D-19-35285R1). We thank you very much for giving us an opportunity to revise the manuscript. We appreciate editors and reviewers very much for their positive and constructive comments and suggestions. In accordance with the format requirements of PLOS ONE, we have revised the manuscript. The main changes are as follows:

A. Corrected the manuscript again, including words and usage, to make it clear, correct, and unambiguous. We also unify the definition of age groups.

B. Revised the format of the manuscript again. The first page contains the title and authors information merely. 

C. The differences among Japanese Encephalitis cases were analyzed again by SPSS software (version 20; IBM Corp., Armonk, New York). We also added SPSS software information to the manuscript. The statistical analysis results were submitted as other supporting information.

D. The uploaded figures were re-plotted in accordance with the publishing requirements, and the format of figures is TIFF, 600 DPI. All figures in the manuscript had been corrected by PACE.

#Response to reviewer 2 

Reviewer #2: The authors have addressed my concerns regarding grammatical errors from the previous draft. This revised draft is much improved and easier to follow. I only have a few minor editing suggestions listed below.

Response: Thank you for the helpful comments, which have helped us a lot. In accordance with the comment, we have carefully revised the manuscript. In addition, to make sure the statistical analysis been performed appropriately and rigorously, the statistical analysis results were submitted as other supporting information. Thanks for your patient work again.

Has the statistical analysis been performed appropriately and rigorously? 

Reviewer #2: I Don't Know

Response: The differences among Japanese Encephalitis cases were analyzed again by SPSS software (version 20; IBM Corp., Armonk, New York). The statistical analysis results were submitted as other supporting information.

Line 41 a high-high cluster? Should this be “a high cluster characteristic?”

Response: It is the definition of local indicators of spatial association (LISA), when the Z-score was > 1.96 (P�0.05), the distribution of JE cases was clustered, which indicated that the surrounding features had similar values (high-high or low-low).

Line 74 Can you explain non-vector transmission or give an example here?

Response: We are also concerned about the conclusion of vector-free transmission of JE virus, however, this reference entitled "Vector-free transmission and persistence of Japanese encephalitis virus in pigs" (doi: 10.1038/ncomms10832 PMID: 26902924) had described the possibility. They demonstrate that JEV can be transmitted between pigs in the absence of arthropod vectors. To keep the original meaning, we change "no vector" to "vector-free".

Line 90 Change provide to province.

Response: Change provide to province. We also recheck similar errors according to the full text.

Lines 105-106 This sentence seems redundant and should be removed, if I am understanding it correctly: “Private information had been completely erased from the original JE dataset; therefore, the data we used for this analyses no longer included names, addresses, etc.” Lines 100-101 already state that “Personal identifying factors, such as name, address and clinical information, were removed.”

Response: Lines 105-106 those sentences indeed redundant as pointed out, we deleted them. In addition, not only lines 105-106 seem redundant but also lines 107-108, however, to stress the protection of personal information, we remain lines 107-108.

Line 109 What software? ArcGIS?

Response: Add "ArcGIS" in line 109. The complete sentence is “All data were analyzed by the ArcGIS software (version 9.3; ESRI, Redlands, CA) and SPSS software (version 20; IBM Corp., Armonk, New York) automatically.”

Line 123 Change revealed to displayed.

Response: Change revealed to displayed.

Lines 125-126 Edit sentence to: A local Moran’s I coefficient >0 indicates that the data have a positive spatial correlation.

Response: Edit sentence to “A local Moran’s I coefficient >0 indicates that the data have a positive spatial correlation.”

Line 128 Edit sentence to: In addition, the Z-score and P-value were utilized for data analysis.

Response: Change Meanwhile to In addition.

Line 212 Edit sentence to: Recently, there has been a great amount of support…

Response: Edit the sentence to "Recently, there has been a great amount of support for vaccinating adults against JE."

Line 213 Change quantity to number.

Response: Change quantity to number.

Line 218 Change “respond to” to improve.

Response: We carefully analyzed these words, such as improve, reflect, and suitable for, however, we believe that the word "respond to" is clear to show the meaning we want to say.

Line 219 Edit sentence to: Although the incidence and mortality of JE have shown a downward trend in Yunnan Province…

Response: Change had to have shown.

Line 222 Remove: …which seems to be a downward trend of incidence and mortality. This is redundant.

Response: Delete this sentence.

Line 229 Edit sentence to: In a study that was reported in 2009, the infection rate of Culex tritaeniorhynchus was 13.2% [31], indicating a high level of virus activity.

Response: Edit sentence to “In a study that was reported in 2009, the infection rate of Culex tritaeniorhynchus was 13.2% [31], indicating a high level of virus activity.”

Line 231 Edit sentence to: In Tibet, during a JE epidemic, the overall seroprevalence…

Response: Edit sentence to “In Tibet, during a JE epidemic, the overall seroprevalence of JEV IgM in pigs was 5.1% [32].”

Line 232 Edit sentence to: Furthermore, JE epidemics have occurred along the border…

Response: Edit sentence to “Furthermore, JE epidemics have occurred along the border of Yunnan Province [33,34]. ”

Line 238 Edit sentence to: This study suggested that the risk of another JE epidemic occurring is still high and that prevention efforts in Yunnan Province should not be neglected.

Response: Edit sentence to “This study suggested that the risk of another JE epidemic occurring is still high and that prevention efforts in Yunnan Province should not be neglected.”

Line 242 Change any to human.

Response: Change any to human.

Line 248 Change “caused or infected” to acquired.

Response: Change “caused or infected” to acquired.

Line 249 Change discovery to diagnostic testing.

Response: Change discovery to diagnostic testing.

Line 257 Change has to have.

Response: Change has to have.

Other revisions：

A. We unify the definition of age groups. The similar various expression, such as less than 10 years old, younger 10 years old, under 10 years old, and≤ 10 years were unified with equal or less than 10 years old, abbreviated as LEQ10; in addition, these versions of age groups, such as older than 10 years, greater than 10 years, above 10 years, and>10 years were unified with greater than 10 years, abbreviated as GTR10. We re-plotted the figures based on the revisions.

B. Delete the line 244 word “Although”. Edit line 244 sentence to “The JE cases were not randomly distributed [36], they also show the same characteristics in Yunnan Province, especially in the areas of Jinghong City, Guangnan County, Yanshan County, Funing County, and Mengzi City, which are hot spots for the JE epidemic.”

C. Change pmid to PMID in references.

---

## [Editor Report · Decision Letter 2]

30 Mar 2020

The spatiotemporal distribution of Japanese Encephalitis cases in Yunnan Province, China, from 2007 to 2017

PONE-D-19-35285R2

Dear Dr. Mao,

We are pleased to inform you that your manuscript has been judged scientifically suitable for publication and will be formally accepted for publication once it complies with all outstanding technical requirements.

With kind regards,

Tian Wang, PhD

Academic Editor

PLOS ONE
---

## [Editor Report · Acceptance letter]

1 Apr 2020

PONE-D-19-35285R2 

The spatiotemporal distribution of Japanese Encephalitis cases in Yunnan Province, China, from 2007 to 2017 

Dear Dr. Mao:

I am pleased to inform you that your manuscript has been deemed suitable for publication in PLOS ONE. Congratulations! Your manuscript is now with our production department. 

With kind regards,

on behalf of

Dr. Tian Wang 

Academic Editor

PLOS ONE